# Indigenous Protected and Conserved Areas (IPCAs), Aichi Target 11 and Canada's Pathway to Target 1: Focusing Conservation on Reconciliation

**Melanie Zurba** [1,2,*], **Karen F. Beazley** [1], **Emilie English** [1] and **Johanna Buchmann-Duck** [1]

1    School for Resource and Environmental Studies, Dalhousie University, P.O. BOX 15000, Halifax, NS B3H 4R2, Canada; Karen.Beazley@Dal.ca (K.F.B.); Emilie.English@Dal.ca (E.E.); jbuchmannduck@dal.ca (J.B.-D.)
2    College of Sustainability, Dalhousie University, P.O. BOX 15000, Halifax, NS B3H 4R2, Canada
*    Correspondence: Melanie.Zurba@Dal.ca; Tel.: +1-902-494-2966

**Abstract:** This article provides analysis of the issues relating to movement towards new models for Indigenous-led conservation in light of Canada's initiatives for greater protected areas representation through Target 1. We provide a background on Canada's Pathway to Target 1, which is based on Target 11 from the Aichi Biodiversity Targets set forth by the Convention on Biological Diversity (CBD). We contemplate the past, present and future of colonization and reconciliation in Canada, and consider the influence of international declarations, programs and initiatives on the potential for the formation of Indigenous Protected and Conserved Areas (IPCAs). We then provide an analysis of "wicked problems" that Indigenous communities, governments, and other stakeholders in protected areas will need to navigate towards implementing the IPCA approach in Canada. We outline the different types of Indigenous involvement in protected areas and how they potentially fit within the principles for the development of IPCAs. We then turn our discussion to the need to refocus conservation on reconciliation by restoring nation-to-nation relationships and relationships between the land and peoples. The lessons we draw have potential parallels for other nation states, particularly those signatory to the CBD and with a colonial history, aiming for biodiversity conservation and reconciliation with Indigenous peoples through IPCAs.

**Keywords:** biodiversity; conservation targets; protected areas; Indigenous peoples; IPCAs; reconciliation; Aichi Biodiversity Targets

## 1. Introduction

Events over the last two decades have significantly changed the landscape for conservation, protected areas, biodiversity protection, as well as the individual and collective rights of Indigenous peoples. One of the markers of this significant shift in the conservation paradigm is the Durban Accord, which was adopted by the International Union for the Conservation (IUCN) at the V[th] IUCN World Parks Congress in Durban South Africa in 2005. The IUCN is the world's largest conservation organization, bringing together 1300 member organizations, including governments, non-governmental organizations (NGOs) and communities from across the globe to address the most pressing conservation issues [1]. The Durban Accord represents a historical moment in conservation and the human rights movement and identifies a "new protected areas paradigm", which not only rejects many long-held assumptions, policies and practices, but perhaps more importantly proposes an entirely different method for establishing, governing, and managing national parks and protected areas [2]. This new paradigm directly confronts and works to decolonize colonial conservation [2]:

> In this changing world, we need a fresh and innovative approach to protected areas and their role in broader conservation and development agendas. This approach demands the maintenance and enhancement of our core conservation goals, equitably integrating them with the interests of all affected people. In this way, the synergy between conservation, the maintenance of life support systems and sustainable development is forged. We see protected areas as vital means to achieve this synergy efficiently and cost-effectively. We see protected areas as providers of benefits beyond boundaries—beyond their boundaries on a map, beyond the boundaries of nation states, across societies, genders and generations. [3] (p. 220)

Coinciding with the Durban Accord, new approaches to governance of protected areas have been emerging globally and have brought greater equity into governance systems with mixed outcomes. Such approaches have been vexed by a variety of issues relating, but not limited to, power sharing, capacity building, and navigating legal frameworks and jurisdictions [4]. Concurrent with the challenges of bringing greater equity into protected areas' governance systems are the challenges of developing new strategies for protecting the planet's unprecedented losses of biodiversity [5]. In 2010, the Tenth meeting of the Conference of the Parties to the Convention on Biological Diversity (CBD) adopted a revised *Strategic Plan for Biodiversity*, which included the launching of the Aichi Biodiversity Targets [6]. The Aichi Targets, as they are commonly known, include 20 individual targets, which are meant to be achieved by the parties of the CBD by 2020. In a delayed response to the Aichi Targets, Canada established a series of national goals known as "The 2020 Biodiversity Goals and Targets for Canada" [7]. Aichi Target 11[1] is partially reflected within Canada Target 1, which states:

> By 2020, at least 17% of terrestrial areas and inland water, and 10% of marine and coastal areas of Canada are conserved through networks of protected areas and other effective area-based measures. [7] (p. 23)

In addition to attempting to advance Target 1, the Government of Canada is also attempting to move forward with reconciliation efforts and putting into practice the Truth and Reconciliation Commission's Calls to Action (described in greater detail in Section 3.1) as well as the United Nations Declaration on the Rights of Indigenous Peoples (UNDRIP). In 2017, the Indigenous Circle of Experts (ICE) was invited, created and their collective work commemorated by national Indigenous organizations and federal, provincial and territorial jurisdictions engaged in the Pathway to Target 1 [8,9]. The members of the ICE were tasked with examining how Canada Target 1 and consequently Canada's global commitment to the CBD could be met in an equitable manner, including the development of Indigenous-led conservation, which ICE would come to call "Indigenous Protected and Conserved Areas (IPCAs)" [8] (p. iii). In the Spring of 2018, the ICE released a report entitled *We Rise Together: Achieving Pathway to Canada Target 1 through the creation of Indigenous Protected and Conserved Areas in the spirit and practice of reconciliation* [9].

Our paper reflects on the *We Rise Together* report as a monumental shift in Canada's guiding frameworks for protected areas. We explore how the proposed IPCAs relate to international biodiversity conservation targets, Canada's Target 1, the context of complex histories and relationships between Indigenous peoples and the Canadian government, and national and international guidance on protected areas. We provide an analysis of the "wicked problems" that Indigenous communities, governments, and other stakeholders in protected areas will need to navigate towards implementing the Indigenous-led approach in Canada. These issues are of relevance to other nations grappling with implementing Indigenous-led protected areas in the context of international biodiversity conservation

---

[1]   Aichi Target 11 reads, "By 2020, at least 17 per cent of terrestrial and inland water areas and 10 per cent of coastal and marine areas, especially areas of particular importance for biodiversity and ecosystem services, are conserved through effectively and equitably managed, ecologically representative and well-connected systems of protected areas and other effective area-based conservation measures, and integrated into the wider landscape and seascape" [6] (p. 9).

targets, particularly those with a colonial history of dispossession of Indigenous territories and displacement of Indigenous peoples for colonial purposes, including protected areas.

## 2. Protected Areas in Canada, Aichi Biodiversity Targets and Canada's Pathway to Target 1

The Government of Canada has adopted the IUCN definition of a protected area as: "a clearly defined geographical space, recognized, dedicated and managed, through legal or other effective means, to achieve the long-term conservation of nature with associated ecosystem services and cultural values" [7,10]. Once an area has been identified as a protected area, one of the IUCN's seven protected area management categories may be applied to further define the parameters of protection and use for the area [10]. The wording "legal or other effective means" is important for the establishment of IPCAs in Canada because it provides a mechanism through which IPCAs can be recognized and reported without being co-opted by traditional colonial models for protected areas, which is an important aspect of self-determination [11]. The international policy discourse, put forth by the IUCN, clarifies that "other effective means" of conservation can refer to "recognized traditional rules" [10].

Language exists to describe what "other effective area-based conservation measures" (OECMs) entail in the context of Aichi Target 11 and thus of relevance to Canada target 1, and draft guidelines have been released by the IUCN-World Commission on Protected Areas (WCPA) [12]. The inclusion of OECMs in Aichi Target 11 acknowledges that " . . . areas outside the recognized protected area networks also contribute to the effective in-situ conservation of biodiversity . . . . [including] territories and areas governed by . . . Indigenous Peoples . . . , and shared governance" [12] (p. 11). Whereas protected areas have conservation as a primary objective, an OECM "should deliver the effective in-situ conservation of biodiversity, *regardless of its objectives*" [12] (p. 14, emphasis added). Protected area practitioners and scholars have expressed concern about the lack of a precise definition for OECMs potentially leading to inconsistent designations, which fall short of conservation objectives [13–17]. For example, Lemieux et al. [15] demonstrated how this concern became a reality in Canada when the Department of Fisheries and Oceans designated marine refuges as OECMs even though these areas are left exposed to external industrial pressures. Accordingly, newly drafted international guidelines aim to provide sufficient information for interested parties to apply the OECM concept and to recognize these areas in reporting to the CBD [12]. Such guidance should support good practice in establishing IPCAs in Canada, regardless of their primary objectives, and in recognizing Indigenous peoples, territories and governments and their roles, responsibilities and contributions to effective in-situ conservation of biodiversity.

The prominence of Aichi Target 11 has resulted in Canada's establishment of Target 1, which initiates the race to meet area-based quantitative targets by 2020, with a risk of insufficient consideration of qualitative aspects [14,15], including those relevant to IPCAs. MacKinnon et al. (2015) [14] pointed out that, in *Canada's Fifth Report to the CBD,* the qualitative aspects of Target 11 are not evaluated, and thus there is a sole focus on the quantitative aspects. While establishing greater area of protection is important, emphasis on an area-based focus has resulted in a proliferation of parks that lack demonstrable conservation impact and leave many questions around how Indigenous peoples might play meaningful roles in the future of protected areas [15–17]. Canada's focus on Aichi Target 11, has also led to the neglect of other Aichi Targets that could serve as important guidelines for structural changes leading to meaningful Indigenous involvement and leadership in protected areas. Of the 20 Aichi Targets, Aichi Targets 14 and 18 speak the most directly to Indigenous peoples. Target 14 states:

> By 2020, ecosystems that provide essential services, including services related to water, and contribute to health, livelihoods and well-being, are restored and safeguarded, taking into account the needs of women, indigenous and local communities, and the poor and vulnerable. [6] (p. 9)

Target 18 also has significant implications for Indigenous peoples and reconciliation in Canada:

By 2020, the traditional knowledge, innovations and practices of indigenous and local communities relevant for the conservation and sustainable use of biodiversity, and their customary use of biological resources, are respected, subject to national legislation and relevant international obligations, and fully integrated and reflected in the implementation of the Convention with the full and effective participation of indigenous and local communities, at all relevant levels. [6] (p. 9)

Collectively Targets 14 and 18 have significant implications for Indigenous peoples and reconciliation in Canada. Both international guidance [18,19] and the ICE report clearly state that IPCAs should be Indigenous-led, and as such IPCAs should allow for the integration of traditional knowledge in defining conservation and sustainable use for those communities. Thus, it should be called into question whether the "equitable" management goal within Target 11 could be achieved in IPCAs without the inclusion of Targets 14 and 18. To support the achievement of multiple targets, it will be important that indicators for each be developed and equally considered. In their evaluation of Aichi Target progress, Tittensor et al. [20] noted that there are no identifiable indicators for Target 18, suggesting an urgent need for the development and integration of indicators across targets.

## 3. Past, Present and Future: Canada's Colonial History and Guiding Frameworks for Indigenous Protected and Conserved Areas (IPCAs)

The increasing prominence of Indigenous-led protected areas, combined with heightened discourse and action on reconciliation efforts in Canada, presents an opportunity to achieve conservation and reconciliation concurrently. This explicit connection between the status of Indigenous peoples' culture and nature is clearly acknowledged in *We Rise Together*. As Eli Enns, Co-Chair of the ICE, states, "Whenever you find intact ecological biodiversity, you find intact, thriving, cultural holistic diversity" [9] (p. 73). Similarly, Niigaan Sinclair, Anishinaabe professor of Native Studies at the University of Manitoba, observes, "You can't have a bear clan with no bears. Indigenous nations know this. It's called diversity. Canada's built on it. We just need to remember who gave us it" [21] (n.p.).

Modern frameworks for IPCAs demonstrate due consideration of important facets of reconciliation (such as Indigenous-led), which, if applied correctly, may serve as a pathway to reparation and decolonization of peoples and nature. As recognized by the Truth and Reconciliation Commission of Canada (TRC) [22], acknowledging the truth is an important precursor to reconciliation, and thus examining Canada's colonial past and present is a critical component of moving forward with IPCAs.

### 3.1. Canada's Protected Areas: The Colonial Past and Present

Protected areas, as embodied in Canadian law and imagination, have been developed through a paradigm which holds paramount the preservation of a people-free "wilderness". National parks in Canada were early tools that deliberately perpetuated colonial injustices and continue to operate as such today despite some recent improvements [9,23]. Many well-established and world-renowned national parks in Canada, such as Banff, Jasper, and Riding Mountain National Parks, as well as provincial parks such as Quetico in Ontario, were founded on and continue to struggle with a legacy of colonial dispossession of Indigenous peoples [23–26]. The creation of these defined regions, which continue to act as conservation enclosures, not only intentionally removed Indigenous peoples from their lands and dispossessed them of their territories, but also perpetuated the illusion of "wilderness" as being pure and devoid of human life and influence [27]. This "wilderness" model of national parks concurrently advanced capitalist enterprises such as sport hunting, recreation, and tourism, while excluding Indigenous peoples as beneficiaries in any capacity or form [9,23]. It became an exercise of "primitive accumulation", entailing both the dispossession of land and the enclosure of a commons in favor of specific interests [28]. The new concept for IPCAs, conversely, is holistic and based on conservation that includes people and culture.

Many Indigenous peoples distrust the concept and nature of protected areas in Canada. Marylyn Baptiste (former Chief of the Xeni Gwet'in) stated that "since my dad was chief, as far as I've always learned, governments in Canada have reserved parks for their own benefit for later use", and cites Bill F4 in BC which proposed opening parks to mining exploration [29] (p. 20). Other objectives, such as species protection, can prompt conservation efforts from the government that perpetuate the exclusionary colonial model of conservation and wilderness. For example, a caribou range plan proposed by the Government of Alberta was not well received by Indigenous peoples. They believed that the plan was spurred by the province's desire to turn their traditional territory into parkland and suppress Indigenous land rights, rather than efforts to protect the caribou herds [30]. However, alternative perspectives amongst Indigenous leaders exist in favor of protected areas. As John Amagoalik, Inuit Elder and Statesman, stated, "The Government of Canada needs to fulfill its overdue promise to create a national marine conservation area in the Landcaster Sound. Inuit rely on the abundance of Tallurutiup Tariunga and expect it to be protected" [31]. In December 2018, the negotiation of the Tallurutiup Imanga Inuit Impact and Benefit Agreement (IIBA) was announced, and once approved by the Canadian government, this will complete "the largest unified land and water protected area in Canada" [32] (n.p.).

While the colonial model of protected areas persists, there is an increasing acknowledgement within Canadian parks institutions (federal, provincial, and territorial) of the "history of exclusion" [9], and in some cases attempts at reparations are being made [23]. As stated by a participant in Exploring Empowerment for Indigenous Protected and Conserved Areas in B.C., a workshop held by the David Suzuki Foundation, "reconciliation is also restitution—returning things taken" [33] (p. 17). Furthermore, Canadian governments have a fiduciary responsibility to consult and accommodate Indigenous peoples when establishing a new protected area, according to Section 35 of the Constitution Act, 1982.[2] Significant policy and legislative changes occurring over the past few decades are beginning to affirm the special relationship Indigenous peoples have with lands, including those within national parks [36]. Nevertheless, Indigenous and legal scholars agree that many restitutions and reparations for communities and structural changes to protected areas institutions will be essential if protected areas are to become inclusive of Indigenous values, aspirations and knowledge systems [36–39]. Furthermore, fundamental considerations relating to territory and sovereignty are due when considering parks and protected areas. It is imperative for Canada, as well as provincial and territorial jurisdictions, to acknowledge the relationship and continuing ownership that Indigenous peoples have with their traditional territories if protected-area systems are to truly move to a stage of decolonization [40]. Connecting to this imperative are state- and community-driven processes of reconciliation. Both processes include acknowledging the need for reparations for historical wrongdoings, as well as the development of new pathways for relationships rooted in truth, justice, and healing [22]. However, several critiques of state-driven processes exist, including that formal reconciliation processes have been used to divert attention away from legal restitutions and disputes over lands and resources [37]. Despite such critiques, it is important to understand the context of formal reconciliation processes in Canada because they will continue to shape the development of policy affecting Indigenous peoples and their lands.

As a first step towards reconciliation, the Government of Canada established the Truth and Reconciliation Commission of Canada in 2008 as a part of the Indian Residential Schools Settlement

---

[2] Canada's supreme law is written in the Constitution Acts of 1867 and 1982. The Constitution Acts outline Canada's system of government (as a federation), as well as the civil rights of all Canadians and those in Canada. According to section 91.24 in the Constitution Act of 1867, the federal government has jurisdiction over "Indians, and Lands reserved for Indians". Through section 35 of the Constitution Act 1982, "the existing Aboriginal and Treaty rights of Aboriginal people in Canada are hereby affirmed"; the Act also clarifies that "Aboriginal Peoples of Canada include the Indian, Inuit, and Métis Peoples of Canada" [34,35].

Agreement.[3]  Following the end of the hearings involving testimonials from Residential School survivors in 2015, the TRC developed a set of "Calls to Action" that would serve as a guide to the processes of reconciliation for the Government of Canada and all Canadians [22]. The need to acknowledge Indigenous lands is referred to several times in the TRC's Calls to Action. In addition to the Calls to Action, the TRC outlines "principles of reconciliation", which include recognizing it as a process of " . . . healing relationships through truth sharing, . . . and redress [for] past harms", as well as " . . . constructive action on addressing the ongoing legacy of colonialism . . . " [42] (p. 3). The principles of reconciliation can be interpreted in many different ways, according to each (Indigenous) nation's understanding of what it should encompass [22]. Some factors that may pose challenges to reconciliation are deeply entrenched in the structural roots of the nation-state known as Canada, and will require long-term political will if they are to be gradually overcome. Encouragingly, some solutions and models for reconciliation can be found by looking to the roots of Indigenous-settler relations, as in the case of the nation-to-nation spirit of many treaties.[4]

While broader discussion about sovereignty, treaties, UNDRIP, Indigenous law, Aboriginal law, and Canadian law are beyond the scope of this paper, it is important to acknowledge the weight of these issues as they are at the root of the set of problems and opportunities that reconciliation is attempting to address. In *We Rise Together*, the ICE states that reconciliation "means identifying the appropriate healing process for restoring relationships: first, between Crown and Indigenous Peoples, recognizing what has not worked in the past so it is corrected moving forward in the spirit of peace and friendship; and second, between all people (Indigenous and non-Indigenous) and the lands" [9] (p. 7). Creating an appropriate forum for honest discussion around targets and IPCAs is important, and will likely need to happen within the context of "ethical space" as indicated in *We Rise Together* [9]. The concept of ethical space includes characteristics such as the creation "of a place for knowledge systems to interact with mutual respect, kindness, generosity, and other basic values and principles" and is "a space for collaboration and achieving common ground" [9] (p. 17).

*3.2. Guiding Frameworks for Indigenous Protected and Conserved Areas (IPCAs)*

The IUCN, as the leading global conservation organization, brings governments, non-governmental organizations (NGOs), the United Nations (UN), and communities together "to forge and implement solutions to environmental challenges," including those related to protected areas governance through various programs and initiatives [1] (para. 3). Since the Durban Accord, there has been increasing participation of Indigenous peoples and Indigenous Peoples' Organizations (IPOs) in developing global protected area (and other forms of environmental) governance frameworks through IUCN programs and initiatives. The IUCN Programme on Protected Areas administers standards for global protected areas such as the Green List of Protected and Conserved Areas, which sets "the new global standard for protected areas in the 21st Century" [44] (para. 2). Within the guiding policy document, Green List of Protected and Conserved Areas: Standards, Version 1.1, under criterion for "Good governance" of protected areas, is the indicator that states: "The site's local governance structures and mechanisms recognise the legitimate rights of Indigenous Peoples and local communities" [45] (p. 13). There is also reference in the standards to the need to consider Indigenous Community Conserved Areas and OECMs under "Sound Design and Planning" of protected area systems [45]. Several other programs and initiatives focus on developing frameworks for Indigenous participation in protected areas, as well as specialized working groups. In particular, the Indigenous

---

[3]  Indian residential schools were first established in Canada in the 1880s, with the last one closing in 1996. The government's intention with this system was to "remove the Indian from the child" by forcibly removing them from their families and communities, and forbidding the use of Indigenous languages and any expressions of Indigenous cultures [41].

[4]  Treaties, as defined by the Government of Canada [43], are "agreements made between the Government of Canada, Indigenous groups, and often provinces and territories that define ongoing rights and obligations on all sides". Both historical (70 treaties between 1701 and 1923) and modern-day treaties (25 treaties since 1971) are relevant in the context of protected areas.

and Community Conserved Areas (ICCA) Consortium [18] has been highly influential in advancing frameworks for Indigenous rights with regards to protected areas.

The ICCA Consortium is a "membership-based civil society organization" that was formed in 2010 with the mission "to promote the appropriate recognition of, and support to, Indigenous peoples' and community conserved areas and territories (ICCAs) at local, national and international levels" [46] (para. 1). At an international level, the ICCA Consortium outlines three defining characteristics of ICCAs, which are echoed in the IUCN Protected Area Guidelines. These are that the community has a close and meaningful relationship to the area; the community is the primary decision maker for site management and has the power to develop and enforce regulations; and management by the community leads to conservation regardless of motivation [46,47]. Similarly, the ICE (2018) states three defining elements of IPCAs for the Canadian context: "they are Indigenous-led; they represent a long-term commitment to conservation; and they elevate Indigenous rights and responsibilities" [9] (p. 5). The concept of "Indigenous-led" forms one of three essential elements of an IPCA in both the ICE [9] and the ICCA Consortium [18] characterizations.

Given the history of colonization in Canada, Indigenous-led is a critical element to ensuring reconciliation is a part of the IPCA process. The ICE defines Indigenous-led as where "Indigenous governments have the primary role in determining the objectives, boundaries, management plans and governance structures for IPCAs as part of their exercise of self-determination" [9] (p. 36). The concept is also being used in environmental assessment and has been defined as being on the terms of Indigenous people, with their approval, and where Indigenous peoples "are involved in the scoping, data collection, assessment, management planning, and decision-making about a project" [48] (p. 10). These definitions and characteristics of Indigenous-led are important to keep in mind in discussions of how IPCAs relate to Canada's conservation targets. Importantly, an IPCA should not be evaluated outside of the Indigenous community's concerns or by the expectations of external actors [47]. Steven Nitah, negotiator and former Chief of the Lutsel K'e Dene First Nation, described this imperative for self-determination and effective partnerships:

> Fundamentally, the starting place for an (Indigenous Protected Area) IPA must be self-determination by the [I]ndigenous peoples themselves, but once declared, IPAs become the basis for building effective partnerships between [I]ndigenous and public governments and other entities, including NGOs, research institutions, and the philanthropic community. [49] (p. 4)

The consideration of Indigenous-led is critical in light of the Aichi Targets in Canada, as well as elsewhere. These targets have created a potentially problematic set of standards, by which IPCAs may be evaluated by criteria that have been defined outside of the Indigenous community, and the impetus for IPCA establishment and national/international reporting may be imposed by the nation state, rather than "led" by the Indigenous community.

In Canada, management arrangements with the federal, provincial, or territorial governments that, while highlighted as Indigenous-led, may be worth reexamining as part of the reconciliation process. Partnerships should be considered Indigenous-led where (and only where) the choice to partner with the government comes from the Indigenous community or Indigenous government itself and not from the federal, provincial, or territorial government [29]. An example of this is Dinàgà Wek'èhodì Candidate Area in the Northwest Territories where, after proposing that an area of ecological and cultural importance be recognized, the Tłı̨chǫ Government (an Indigenous self-government in the Northwest Territories) put forward a proposal to protect the area using territorial legislation, which they are developing in partnership with the Government of the Northwest Territories [50,51]. Recently, the Dehcho First Nations and the Government of Canada announced their collaborative effort to establish an IPCA in traditional Dehcho territory, called Edéhezíe Protected Area [52,53]. For Canada, this is the first officially announced IPCA [53].

The Government of Canada finalized an agreement with the Dehcho First Nations in association with the designation of Edéhezíe Protected Area as a National Wildlife Area by 2020 [52,54]. Dahti

Tsetso, Resource Management Coordinator for the Dehcho First Nation, stated that the agreement " . . . will give us some capacity to start addressing the goals of our communities and approaching protection in ways that make sense to them, that helps our communities approach stewardship in a meaningful way" [55] (para. 10). Under this arrangement, Dehcho guardians would clearly be important in monitoring and management, but it may be unclear how much control they have had in establishment of the area. Designating an IPCA under a pre-existing federal protected area category may present issues and risk an imbalance of power with more resting in the federal government's hands. Up until the designation process, however, this initiative appears to have been Indigenous-led: the Dehcho First Nations and Tłıchǫ government requested in June of 2010 that Edéhezíe be designated as a National Wildlife Area [52]; and, in 2018, the Dehcho First Nations (DFN) Annual Assembly,

> . . . resolved that the Assembly: 1. approves and enacts the Dehcho Protected Area law (2018); 2. authorizes the DFN Grand Chief to enter into the Edéhezíe Establishment Agreement with Canada on behalf of the Dehcho First Nations; and 3. authorizes the Dehcho First Nations to finalize the Establishment agreement with Canada and to do such other things as may be necessary to permanently protect Edéhezíe . . . [56] (p. 2)

The Dehcho First Nations community has been pushing for protection of their cultural lands for years considering the threat of mining and other resource development in the area [55]. Indigenous leadership in this case is clear; however, it is also worth questioning if this request was made because of inadequacy of the legislative tools that are available to protect the region from industrial development, and what this could mean for further interference from the federal government. Although the Dehcho are finalizing an establishment agreement with Canada, it remains an open question as to whether, over time, the power imbalance of a colonial designation will or will not be redressed, even though the process has been Indigenous-led, and with an agreement in place for Indigenous management.

## 4. Potential Types of Indigenous Protected and Conserved Areas (IPCAs) in Canada

IPCAs have the potential to fit into Target 1 as either a protected area or an OECM depending on whether it meets the definition and criteria, including among others its objectives for conservation. However, despite such categorizations, Indigenous peoples have been managers of their lands and resources since time immemorial [13]. While the decision to establish an IPCA must come from Indigenous leadership, a range of partnerships with government or other outside parties may be appropriate [9]. Through their processes of engagement, the ICE [9] concluded that the type of partnerships and the degree to which decision-making authority is shared should be determined by the Indigenous group based on the objectives and needs of their nation or government. Within Canada, Indigenous peoples have been involved in co-management and joint management agreements for several decades, and have also created independent governance models to protect their lands and culture from harms arising from industrial and governmental pressures [9,57]. The different types of Indigenous involvement in protected areas are outlined in Table 1, which also provides examples and information on the traditional territory and jurisdiction(s). Outlining the different potential types and examples of IPCAs provides insights into how IPCAs might be categorized and defined according to Canada's Pathway to Target 1. Another important consideration is whether the Indigenous community wants to have their involvement in protected areas counted at all. In the future, such processes should be initiated only where the Indigenous government has indicated a distinct interest.

**Table 1.** Indigenous involvement in protected areas management in Canada. Models adapted from *We Rise Together* by the ICE [9].

| Type of Indigenous Involvement in Protected Area | Example | Traditional Territory/Province/Territory | Jurisdiction/Decision-Making Authority |
|---|---|---|---|
| Advisory board | Fundy National Park | Mi'kmaq/Wolastoqiyik/New Brunswick [58] | Federal authority |
| Joint or cooperative management | Torngat Mountains National Park | Inuit from Nunavik/Inuit from Nunatisavut/Newfoundland and Labrador [59] | Federal authority |
| Conservancy | Bear Island Conservancy | Nat'oot'en Nation/British Columbia [60] | Shared governance (with the province) |
| Tribal park | Tla-o-qui-aht Tribal Parks | Tla-o-qui-aht First Nation/British Columbia [61] | Provincial authority |
| Indigenous management | Wehexlaxodıale | Tłıchǫ First Nation/Northwest Territories [62] | Indigenous governance |
| Indigenous governance | Edéhezíe Protected Area | Dehcho First Nation/Northwest Territories [53] | Indigenous governance |

With the ICE's defining qualities of IPCAs in mind (i.e., the process is Indigenous-led; there is a long-term commitment to conservation; and they are in support of Indigenous rights and responsibilities), the designations in Table 1 that would qualify as IPCAs are those that are "conservancies", "Tribal Parks", "Indigenous management", and "Indigenous governance". Unlike Tribal Parks and Indigenous management, conservancies are recognized and defined under provincial law. This means that all conservancies in British Columbia will meet the same criteria: they meet the IUCN definition of a protected area as they are geographically defined, legally protected under provincial law, and are set aside for the protection of biodiversity as well as cultural values of Indigenous peoples [60]. Conservancies allow for the development of natural resources if it is consistent with the protection of biodiversity, uses by Indigenous peoples, and recreational values [60]. However, conservancies cannot be considered "Indigenous-led", as the province still maintains the power to approve decisions. While joint authority might support "Indigenous rights and responsibilities" and show a "long-term commitment to conservation", it cannot be assumed that the process is adequately Indigenous-led. Such adequacy can only be determined by the Indigenous group that is part of the agreement and involved in decision-making.

Tribal Parks are not officially recognized by the Government of Canada, which has the potential to conflict with the need to support Indigenous rights and responsibilities. Tla-o-qui-aht Tribal Parks is a well-established model of a Tribal Park located in British Columbia. These parks are geographically defined, and while Tla-o-qui-aht law may not be officially recognized in Canada [9], the Tribal Park can still meet the protected area definition of "other effective means" noted earlier in this paper. These Tribal Parks are intended to focus on the integration of human wellbeing with healthy ecosystems [61]. Tribal Parks are currently not considered under Canada Target 1 and are not fully recognized as a protected area by the provincial or federal governments [9]. It is important to note that, should government recognition and support come in future, it would need to be the Tla-o-qui-aht leadership that would determine if it ought to be counted as an IPCA under Canada Target 1 as a protected area, OECM designation, or at all.

Indigenous management involves area management that is established and governed by Indigenous peoples without co-management or another type of partnership arrangement, and is a third type of Indigenous management that could fit with the IPCA model. Wehexlaxodıale is an example of Indigenous management, which involves a protected zone established and managed by the Tłıchǫ Government [62]. It is geographically defined and managed under the Tłıchǫ Land Use Plan [62]. Wehexlaxodıale is fully protected for the cultural and heritage value it provides to the Tłıchǫ people. To the Tłıchǫ people, the protection of nature, culture and heritage are intertwined [62]. This differs from typical western notions of conservation that often view nature, culture and heritage as separate aspects of environment and society. Despite this complexity, the Government of the Northwest Territories has reported Wehexlaxodıale as a protected area under "Indigenous Government Administration" on behalf of the Tłıchǫ Government and with their approval. This reporting is to the Conservation Areas Reporting and Tracking System (CARTS) database, which contains all federal, provincial and territorial

data on protected areas [63], and has been used to date by Canada to report progress towards national and international commitments to biodiversity conservation.

The newly established Edéhezíe Protected Area involves a new type of arrangement that is more representative of Indigenous governance. Thus far, of all the types of protected areas involving Indigenous communities, Edéhezíe Protected Area shares the most in common with the ICE's definition for IPCAs. It is yet to be determined if this will give support to the formal establishment of an IPCA category in Canada.

## 5. "Wicked Problems"

Canada has a complex historical, legal, political, and social landscape with several factors that can confound those grappling with whether to recognize IPCAs under Canada's Target 1. These factors can be understood as "wicked problems" [64,65], which are "incomprehensible and resistant to solution" [66]. While innumerous wicked problems could be discussed, we have identified six major types of wicked problems that will need to be addressed before IPCAs can achieve both conservation and reconciliation objectives.

### 5.1. Exclusionary "Wilderness" Paradigm for Protected Areas

As discussed above with relation to the colonial past and present, the "wilderness" paradigm for protected areas, which is based on the nature/culture dichotomy, continues to be pervasive. Concepts of pristine nature and wilderness are often considered central to many conservation approaches, including protected areas designation and management. This paradigm has been underpinned by a static and linear view of nature, the concept of wilderness, and the equation of human presence with ecosystem degradation [27]. Until relatively recently, the underlying assumptions and philosophies guiding the designation of protected areas were not adequately acknowledged or explored [9,67]. Despite the development of new concepts and more inclusionary paradigms, such as IPCAs, which link Indigenous peoples with conservation lands, the dominant exclusionary paradigm has so far proven difficult to supplant, especially when moving from theory to practice [27]. Thus, a profound shift in how the Canadian state, its conservation organizations and its public think about conservation is required to support and align with the new paradigm for protected areas. As is articulated by Steven Nitah,

> I will say that [I]ndigenous contributions have largely gone unrecognized in Canada, in a system that still recognizes only federally, provincially, and territorially legislated protected areas as valid and ignores the fact that for tens of thousands of years our peoples managed the land so well that you thought it was empty. We need to move past those misconceptions and embrace the fact that long before Canadians even knew what a national park was, our peoples were successfully protecting and managing our special places under our own laws and using our own knowledge. [49] (p. 3)

### 5.2. Siloed Colonial Governance

The spaces for interpretation and misinterpretation of existing types of Indigenous-led management discussed in the last section highlight one of the "wicked problems" that can emerge when the dominant Western colonial government aims to categorize Indigenous governance and stewardship. The traditional resource management style of the Canadian government has been to separate natural resources into different management categories, and disconnect them from human wellbeing and cultural continuity [29]. This structure was imposed to "make sense" of a geographically large nation with diverse social, political, economic, ecological, and geological regions [68]. The separation of marine and terrestrial environments, for example, has resulted in different jurisdictions and conservation targets, with significant implications for protected area establishment, reporting and biodiversity outcomes [15], including those important to Indigenous peoples and their interrelated

traditional-territorial lands, waters and lifeways. While structurally dividing the systems responsible for nature and for people's wellbeing reflected the colonial law and approach to governance [69], it has significant incompatibilities with Indigenous governance systems, which tend to be more holistic [70]. The example of Wehexlaxodiale presents a perfect representation of this issue. The intention of the region is to protect cultural values, but in protecting cultural values the Tłı̨chǫ are also protecting nature as these are inseparable concepts.

### 5.3. Variation in Crown–Indigenous Treaties and Land-Claim Agreements

Over a long history of settler–Indigenous relations, a diversity of treaties and land agreements have been entered into by a variety of Indigenous and colonial governments. While much of Canada is covered by historic treaties, most of British Columbia, for example, does not have signed treaties [71]. Several court cases such as *Delgamuukw v. British Columbia 1997 3 S.C.R. 1010* have resulted in decisions that have enabled a modern treaty process that facilitates better Indigenous involvement in decision making [71]. In eastern Canada, many are Treaties of Peace and Friendship in which traditional lands and territories have never been ceded. The ICE in *We Rise Together* notes the diverse nature of these legal agreements and the ability of those communities who did not cede title under treaty to use the court systems to obtain title [9]. Notably, most examples of IPCAs in Canada can be found in BC or the North. Jones et al. [71] argued that in some contexts the reasoning behind this may be due to a lack of certainty around Indigenous land ownership, which has motivated collaboration. Lloyd-Smith [72] stated that the modern land claim agreements in the Arctic present more opportunity for Indigenous peoples to lead conservation and land-use planning initiatives. In contrast, those regions under historic treaty may have limited federal and provincial incentive to enter into truly collaborative models [71]. Even in situations where there may be an appearance of "true" collaboration there may remain a lack of willingness within governments to interfere with Ministerial authority. New models to address this situation are needed, such as the *Northwest Territories Wildlife Act*, which was drafted collaboratively and goes significantly further than other pieces of legislation with respect to collaborative management [73]. For example, Part 2, Section 8 of the *Northwest Territories Wildlife Act* lays out the aim "to promote cooperative and collaborative working relationships for effective wildlife management at the local, regional and territorial levels" [73] (p. 22).

### 5.4. The Non-Devolution of Power by the Government of Canada

In order for IPCAs to be Indigenous-led, there will need to be significant divestments of power made by the Government of Canada. However, power remains highly centralized and the Government of Canada may be reluctant to devolve power to Indigenous governments. Nicol [74] pointed to the dichotomy between the image of a Canadian government that seeks to protect Indigenous rights while at the same time asserting its sovereignty and thereby appropriating Indigenous sovereignty. While power devolution has occurred in the territories to a degree, the rigid bounds of constitutional power in the provinces makes the territorial model of devolution inapplicable [75]. In the recent past, the Canadian government has understood aspects of UNDRIP such as free, prior, and informed consent (FPIC) as a way of undermining state power over decision making [76]. This stance is problematic and demonstrates the unwillingness of the federal government to devolve or share power. Furthermore, if power is fully vested in the current government, then existing governance systems may be vulnerable to changes in government (i.e., through elections or other political means), which may lead to changes in land-based policies [77]. Nicol [76] demonstrated that the Canadian government only values Indigenous rights within the bounds of treaties and land claims, therefore Indigenous rights are constrained to Indigenous lands in the eyes of the state.

### 5.5. The Diverse and Federalist Nature of Canadian Law and Law Making

The constitutional division of powers in Canada between the federal government and provincial and territorial governments adds complexity to the processes by which protected areas become

formalized. Significantly, Indigenous law and Indigenous constitutional orders are not formally recognized under Canadian law. The Canadian constitution was created in a colonial context and in a time when economic gain was the primary objective of resource management, even within parks [78,79]. The division of power between the federal, provincial and territorial governments, as outlined in the *Constitution Act 1867*, further complicates Indigenous–Crown relations [76]. While the creation of Section 35 of the *Constitution Act 1982* and subsequent court cases have begun to transfer some power held by federal, provincial and territorial governments back to Indigenous peoples, these colonial legal structures remain and define environmental decision-making in Canada. For example, Canadian law is structured to recognize written law as superior to oral law, which creates a disadvantage for Indigenous peoples [80]. This relates to issues surrounding sovereignty and the potential for IPCAs to be Indigenous-led, as discussed earlier. This issue also connects to the wicked problem outlined above (Section 5.2), such that a siloed approach to law is also problematic. Curran [69] argued that colonial law operates in silos, with ecosystem and health regarded as separate domains, for example, and, consequently, responsibility is divided by jurisdiction and department. In contrast, responsibilities for the land, health and wellbeing are inseparable in Indigenous laws and lifeways.

*5.6. Reporting*

Monitoring and reporting on implementation and management is a fundamental component of conservation. Conservation reporting under Canada Target 1 thus far has been complex and inconsistent. Each provincial, territorial, or federal jurisdiction self-reports to the Canadian Council on Ecological Areas (CCEA) who tracks the data in their CARTS database [14]. While the CCEA has guidelines for reporting, there is no mandatory requirement for jurisdictions to follow this guidance, and thus, each jurisdiction reports based on its own process and understanding [15]. For example, the CCEA CARTS report from December 2017 shows that Saskatchewan had reported privately protected area while Ontario had not. Furthermore, the Northwest Territories and the Yukon are the only jurisdictions that have reported "Indigenous Government Administration" and these same territories along with Manitoba are the only jurisdictions to report and distinguish government-Indigenous partnership areas [63]. These jurisdictional data are not currently audited for consistency across jurisdictions and are accepted as reported [15].

Ideally, where Indigenous leadership has indicated an interest in reporting IPCAs under Target 1, the community leaders would carry out the reporting processes. However, in Canada, as a result of imbalanced power relationships with other government bodies, many Indigenous governments face capacity issues such as a lack of funding [81]. While reporting is an important part of ensuring conservation objectives are met, it is also critical that the process work to further self-determination. There are currently many structural barriers that stand in the way of Indigenous reporting, including challenges in standardization of reporting, poor recognition of Indigenous governments, and capacity issues. Consistent with international guidance, MacKinnon et al. [14] recommended standardized reporting to avoid inconsistencies across jurisdictions in Canada. This may present a challenge for IPCAs, however, if an Indigenous group must report and demonstrate a certain level of biodiversity protection through an imposed language and understanding of conservation and external methods of practice [13]. Thus far, reporting on protected areas in Canada has only been done through provinces, territories, and federal governments. Ideally, Indigenous governments would be able to control if and how they report their own IPCAs.

*5.7. Tension and Uncertainty between Scientific and Traditional Knowledge in Protected Area Management*

The application of Indigenous knowledge (IK) to protected area management is becoming increasingly prominent in Canada and represents the beginning of a broader acceptance and acknowledgement of various ways of knowing/knowledge systems and worldviews. For example, the Northwest Territories Government has developed a "Traditional Knowledge Policy" to formally recognize the value of these knowledge systems [82]. The ICE [9] represents IPCAs as a manifestation

of Indigenous knowledge systems and states that their promotion in Canada should enhance respect for these knowledge systems. Indigenous knowledge has been used to enhance and complement western science with mixed results [71,83], and pre-conceived notions of what IK is and how it should be incorporated by non-Indigenous partners can present a significant barrier to its application and use [84]. Aichi Target 18 indicates the importance of IK in biodiversity conservation; however, as discussed around reporting, it can also be challenging to apply IK in regions that require external methods of reporting. Houde [85] outlined several challenges such as a lack of confidence in IK by non-Indigenous people, issues around how IK is shared and how the control and ownership of IK might be determined by Indigenous communities, and the complexity of incorporating opposing values into a single framework. Communities face a difficult balance between sharing their knowledge towards asserting self-determination and having their knowledge appropriated by colonial application and resource management initiatives [86]. Elder Albert Marshall articulated the need to redress this dichotomy:

> We need the English language and mainstream knowledge, but at the same time, we want the prerogative of determining for ourselves what we want from the mainstream. We already have the best of who we are from the Ancestors. Let's decide how to put these together in a two-eyed seeing approach. [87] (p. 41)

## 6. Refocusing Conservation for Reconciliation

While frameworks, targets, and broader governmental initiatives may provide the impetus for establishing IPCAs in Canada, it is important to consider that, primarily, these initiatives will require concerted efforts at fundamentally changing the nature of existing relationships. Primarily, restoring nation-to-nation relationships between Indigenous peoples, settlers, and Canadian states is a precondition to just, respectful, and equitable relations. Perhaps less tangibly, but equally important, is creating the space for restoration of the relationship between land and peoples. Whether these are achieved through the establishment of IPCAs or some other means is secondary to the broader objective of using conservation as a tool for reconciliation.

### 6.1. Restoring Nation-to-Nation Relationships

Restoring or (re)building a nation-to-nation relationship is fundamental to reconciliation, and the original treaties may serve as guides on this journey [9,38]. Fundamentally, treaties serve as a moral basis of alliance between Indigenous peoples and settlers, are relationship-based, are intended to be living documents, and secondarily serve as legal documents defining and detailing that relationship [39]. However, it is important to note that treaties were often recorded and interpreted differently by Indigenous nations and subjects of the Crown. The TRC's Call to Action 45 recommends that "the government of Canada jointly develop with Aboriginal peoples a Royal Proclamation of Reconciliation to be issued by the Crown. The proclamation would be built on the Royal Proclamation of 1763 and the Treaty of Niagara 1764, and reaffirm the nation-to-nation relationship between Aboriginal peoples and the Crown" [88] (p. 4). This would be enabled by revisiting the original spirit and intent of the treaties and through enacting a repudiation of concepts (such as the Doctrine of Discovery and terra nullius) used to justify European sovereignty over Indigenous peoples. Other equally important commitments on the part of the Crown are the full implementation of UNDRIP, and commitments to establishing "treaty relationships based on mutual recognition, mutual respect, and shared responsibility for maintaining those relationships in the future" [88] (p. 5). The importance of genuine partnerships underpinned by strong relationships cannot be underestimated. Another important step in rebuilding a nation-to-nation relationship will be to "reconcile Aboriginal and Crown constitutional and legal orders to ensure that Aboriginal peoples are full partners in Confederation, including the recognition and integrations of Indigenous laws and legal traditions in negotiation and implementation processes involving Treaties, lands claims, and other constructive agreements" [88] (p. 5). It can be reasonably asserted that reconciliation efforts that are connected to state-driven

conservation cannot be fully actualized until nation-to-nation relationships grounded in the treaties and important documents like UNDRIP are fully implemented and affirmed.

*6.2. Restoring Relationships with the Land*

Just as it is important to restore a nation-to-nation relationship, institutionally as well as socially, it is equally important to restore, or create the conditions that allow for re-connection between people and the land; for many Indigenous peoples, caring for land is an important part of respecting their cultural responsibilities [89]. As previously discussed, through the creation of protected areas and many other provincial/territorial and federal policies and developments, Indigenous peoples were systematically and forcibly displaced from their territories, consequently loosing access to these sites and their resources, as well as suffering other intergenerational impacts (social, cultural, economic, and spiritual) [9,27]. *We Rise Together* states that the disconnection between Indigenous peoples and their territories "prevented the full functionality of Indigenous legal orders . . . weakened the necessary linkages to inter-generational knowledge transmission and sustainable use . . . " and may have led to the loss of other important cultural and spiritual practices (such as songs, ceremonies, dances, stories, etc.) that were intrinsically tied to the land [9] (p. 28). Indigenous-led conservation efforts and IPCAs enable Indigenous peoples to determine the future of their lands and peoples [89]. A large part of reconciliation through conservation, as well as structural and social reconciliation, is making space for diverse Indigenous knowledge systems, worldviews, laws, etc. This is highlighted in UNDRIP, through which the UN General Assembly recognizes the "urgent need to respect and promote the inherent rights of Indigenous peoples which derive from their political, economic and social structures and from their cultures, spiritual traditions, histories and philosophies, especially their rights to their lands, territories, and resources" [90] (p. 2), and that "respect for Indigenous knowledge, cultures, and traditional practice contributes to the sustainable and equitable development and proper management of the environment" [90] (p. 2). Such considerations will be essential for promoting self-determination, which is integral to reconciliation in the development of meaningful pathways to IPCAs.

## 7. Conclusions

Recent global and national trends towards recognition of Indigenous peoples' leadership in protected areas presents hope for the future of conservation and reconciliation. The spirit and intent of IPCAs reflect the new paradigm and has the potential for positive cultural and social outcomes in addition to the conservation of biodiversity [10]. Countries responding to the new paradigm for biodiversity conservation are now grappling with their colonial histories and modern-day approaches to conservation and are exploring the implementation of protected areas frameworks that are supportive of Indigenous leadership. Each country has their own set of "wicked problems" that they must navigate. Australia has become a model for the shift towards Indigenous leadership in protected areas, boasting health, education, employment and social cohesion outcomes through their Indigenous Protected Area (IPA) program, which is part of the Australian National Reserve System [91]. Australia currently has 75 IPAs over both land and sea jurisdictions, which contribute more than 45% of the National Reserve System's total area [91]. Despite the impressive enhanced representation of protection of biodiversity through the National Reserve System, IPAs continue to be vexed by numerous issues, such as those relating to who has ultimate decision-making authority [92].

IPCAs have the potential to play a major role in meeting many signatory nations' commitments to biodiversity conservation targets under the CBD, along with their associated national targets. Canada's commitments to Aichi Target 11 as reflected in Canada's Target 1 is no exception. Indeed, Canada is in a good position to follow Australia's lead in modeling shifts in IPCA practice both at home and in the global arena. It is important, however, to be mindful of the political motivations behind state support for IPCAs. If the primary objective of government support for IPCAs is to achieve national obligations to meet international conservation commitments and the 2020 target deadline, then this could undermine the meaningfulness of IPCAs and the IPCA process for Indigenous

communities [9,13]. In Canada, the *We Rise Together* report states that the federal government must be willing to support IPCAs regardless of their relationship to conservation targets, as they have a purpose beyond meeting quantitative national objectives [9]. With less than two years left to meet the Aichi Targets, pushing for area-based targets could lead to rushed collaboration and risk inappropriate designations that do not support the objectives of reconciliation or conservation.

Given the "wicked problems" associated with forwarding IPCAs within Canada's Pathway to Target 1, a process that allows for reconciliation through adaptability, capacity building and the strengthening of relationships will be necessary. For example, Jonas et al. [13] argued that, similar to recognition, reporting requires the free prior and informed consent of the Indigenous peoples involved. The ICE [9] identifies some key features necessary for Indigenous peoples to support the shift in paradigm they envision through IPCAs. These include: dedicating sufficient time and resources to explore Indigenous-led conservation and engagement with Indigenous governments; supporting innovative funding models; identifying new partnerships, as well as allies and champions; and creating resources that would support Indigenous governments in their work on IPCAs. It is important to move forward with caution, however, to avoid a situation in which IPCAs become another, hopefully unintended, colonization of Indigenous peoples, their lands and biodiversity to meet Canada's Target 1 for protected areas. This potentiality calls into question the appropriateness of Canada's recognition and reporting of IPCAs if entered into quickly and primarily to aid Canada in meeting its quantitative commitments under the CBD.

Ideally, Indigenous communities would have the ability to define IPCAs according to their particular contexts, thus contributing Indigenous leadership in the process of protected area and OECM recognition. To achieve this, the government could support the "three essential elements" of IPCAs put forward by the ICE (i.e., the process is Indigenous-led; there is a long-term commitment to conservation; and they are in support of Indigenous rights and responsibilities). In so doing, the government would also uphold Indigenous rights and support the self-determination of each community to define their IPCAs. This would also be consistent with IUCN governance guidelines, which note that nations need to be careful not to force top-down management of IPCAs, resulting in the displacement of existing governance structures [47]. As articulated by the ICE [9], close attention will need to be paid to jurisdiction, financial solutions and capacity development, as well as addressing disconnections between—and the dominance of Canadian over—Indigenous worldviews, ontologies, and epistemologies. These factors among others must be considered and overcome in efforts aimed at developing IPCAs as living examples of reconciliation, as well as potential contributions to biodiversity conservation through Pathway to Canada Target 1 and Aichi Target 11.

Considerations around Aichi Targets other than Target 11 are also important. IPCAs inherently offer a commitment to conservation, however they might be better suited to contribute to targets other than Target 11. For example, Aichi Target 18 has a focus on the full and appropriate recognition of Indigenous knowledge systems in conservation, and Target 14 aims to protect ecosystem services and the needs of Indigenous peoples [6]. Accordingly, IPCAs that embed Indigenous knowledge systems and serve to protect ecosystem values of importance to Indigenous communities could contribute to Targets 14 and 18, regardless of whether they fit within the parameters of Aichi Target 11. In these and other cases, such as areas established for cultural regeneration, IPCAs may meet the definition of an OECM and be counted toward Aichi Target 11 or Canada Target 1. Regardless, an Indigenous government's decision about whether they want their IPCA to be counted must be respected by the provincial, territorial, or federal government so as to honor reconciliation processes and FPIC under UNDRIP. Alternatively, an IPCA may have a purpose that does not meet the requirements of a protected area or an OECM under Target 1. In this case, it should still be recognized as a governance model to be supported and protected, as it may serve an important role in reconciliation and conservation processes. It will be critical, however, that colonial governments maintain support for IPCAs, other forms of Indigenous-led conservation and reconciliation without imposing control or regulation.

Even though Canada and other nation states are on a tight timeline to achieve national targets and international biodiversity conservation commitments by 2020, there should also be an understanding that land- and sea-based reconciliation will require due process so that past injustices to Indigenous peoples related to protected areas can be addressed [9]. Inadequately addressing past injustices could create further wrongdoings and reveal Canada's and other countries' interests in establishing IPCAs as opportunistic, rather than being based in genuine efforts to move forward on reconciliation with regards to protected areas. Taking the time to listen to communities and address structural injustices (associated with the wicked problems above) will be important if nation states, including the Government of Canada, are to truly move forward in good faith in a way that acknowledges the intrinsic and holistic value of IPCAs beyond the race to meet targets. As discussed by the ICE [9], IPCAs can be places of cultural regeneration, learning, restoration, and reconciliation, while actively contributing to the protection of biodiversity.

**Author Contributions:** Conceptualization, all authors; methodology, all authors; validation, all authors; formal analysis, all authors; investigation, all authors; resources, E.E. and J.B.-D.; data curation, all authors; writing—original draft preparation, all authors; writing—review and editing, all authors; and supervision, M.Z. and K.F.B.

**Funding:** This research received no external funding.

**Acknowledgments:** We would like to acknowledge Claudia Haas and Lillith Brook of the Government of Northwest Territories, and the reviewers for their helpful feedback.

**Conflicts of Interest:** The authors declare no conflict of interest.

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
