# Peer review of "Indigenous Protected and Conserved Areas (IPCAs), Aichi Target 11 and Canada’s Pathway to Target 1: Focusing Conservation on Reconciliation"

_land, doi:10.3390/land8010010_

Round 1

Reviewer 1 Report

Only one suggestion:

Of course the research focus is about Canada, but as the authors explicit clearly, IPCAs refers to local communities too, and in several countries, mainly in the European Union, are the local communities and their commons the main/ relevant factor in relationship with sustainable land management fact and the dispossession of land in favour of specific interests (as authors recognize in line 148). By this way, perhaps a small paragraph related with the existence of others IPCAs (e.g. in European Union) may be very relevant in order to point out  the importance of this  land management figure, not only in relationship with indigenous people (extremely  importance), but either in relationship with local communities in other countries, because these communities are the only  (and local)  support in order to maintain  an equilibrium  biodiversity/production,  traditional knowledge guard, human-land connexion (lands and rural country alive) maintenance.

Author Response

We sincerely thank the reviewer for spending time with our work. We agree that local community involvement is of high importance when it comes to supporting biodiversity; however, we feel that a discussion on local communities is fairly out of the scope of this paper, which focuses on Indigenous Protected and Conserved Areas. The context for Indigenous peoples is quite different (human rights are very different from Indigenous rights, for example). Indigenous Protected and Conserved Areas in this context do not include local communities. We understand that some international guiding frameworks that apply to Indigenous communities also refer to local communities; however, we feel that discussing local communities would take away from the very important issues faced by Indigenous peoples in Canada and elsewhere.

We agree that greater connection to international context is important and have added language to the abstract, introduction and final section (Conclusions) to situate the research within the international scope for Indigenous protected areas. We conducted a search for sources related to Indigenous PCAs in the European Union but were not able to reveal any.

Reviewer 2 Report

Thank you for asking me to review this manuscript. I am fully behind the authors project and applaud the authors for identifying an exciting moment in Canadian conservation and advancing an analysis of the challenges, or ‘wicked problems’ as they see them.  I do, ofcourse, have some concerns and suggestions for improvement.

1. While I recognize the time constraints inherent in trying to put forward some analysis in a timely manner, I have some concerns about how it is done here. Mostly a purely literature based analysis and review on such a new and emerging process is bound to miss some key points. At the very least, a more thorough review of the popular coverage of Pathway 1 needs to be incorporated, if not interviews with key Indigenous 'insiders'. In an article so clearly aligned with a desire to see Indigenous-led conservation take root in Canada, I would have expected to hear more Indigenous voices. While co-authorship is not always possible, a few key-informant interviews would have added some nuance to the account. Pathway 1 and its implementation is a live act and having someone from 'behind the scenes' would have been beneficial. At the very least, there are quite a few popular pieces containing quotations from members of ICE or key members of the Indigenous leadership on this issue, which should have been at least cited. There is also a fair few allied organizations with important analysis the authors could have built upon. Here are a few I would recommend incorporating (there are others):

Indigenous scholars whose writing would support some of what you put forward (in terms of nation-to-nation relations):

N. Reo, K. Whyte, D. McGregor, M. (Peggy) Smith, and J. Jenkins, “Factors that support

Indigenous involvement in multi-actor environmental stewardship,” AlterNative, vol. 13, no. 2, pp. 58–68, 2017.

Allied scholars involved in supporting ICE:

F. Moola and R. Roth 2018. “Moving beyond colonial conservation models: Indigenous Protected and Conserved Areas offer hope for biodiversity and advancing reconciliation in the Canadian boreal forest.,” Environ. Rev. https://doi.org/10.1139/er-2018-0091

Allied organizations (WCEL) on legal frameworks:

G. Lloyd-Smith, “Indigenous protected areas gaining momentum - but are they recognized by law?",” Environ. Law Alert Blog, 2017.

David Suzuki Foundation – a couple of reports on Tribal Parks and IPCAs in BC

https://davidsuzuki.org/science-learning-centre-article/let-us-teach-you-exploring-empowerment-for-indigenous-protected-and-conserved-areas-in-b-c/

Indigenous leaders, including member of ICE:

V. Courtois and S. Nitah, “Indigenous-led conservation offers a path to global leadership and reconciliation,” The Star, 23-Jan-2018.

E. Enns, “Tribal Parks in Canada: Modern expressions of Indigenous Governance systems,” ICCA Consortium Blog, 2014. [Online]. Available:

https://www.iccaconsortium.org/index.php/2014/10/05/tribal-parks-in-canada-modernexpressions- of-indigenous-governance-systems/.

 2. I also have some concerns about fit and scoping. I am not 100% convinced Land is the best place for this piece. I leave it up to the authors and editors to decide precisely on this issue but let me explain my reasoning. A) I think an allied Canadian audience (such as myself) will enjoy this paper as it discusses a very interesting development in Canadian conservation but it takes too much for granted for a non-Canadian audience or for a Canadian audience who isn’t already fairly well versed in reconciliation, nation-to- nation relationships and the like. This could be resolved with a clearer framing and some footnotes defining terms. What exactly should someone from outside of Canada take away from this? Is there a lesson for other settler colonial states? Is there a lesson about Aichi targets? If the audience is Canadian, perhaps a regional journal would be a better fit. Or perhaps a journal with a readership more familiar with Indigenous people and conservation? B) This is not a research article and contains no data. It is much more of a paper serving to identify research needs or, at least, challenges to actualizing IPCAs in Canada. Perhaps that’s ok for Land, not sure as I am unfamiliar with the journal.

3. I am pleased to see the authors identify some of the risks in Pathway 1 and I agree it is necessary to be a bit cautious about Canada’s motivations – there is a very real possibility that a focus on quantitative targets marginalizes many if not most Indigenous nations and replicates colonial exclusions (had the authors done some interviews, they would have been able to identify a few ways this is already happening.) I agree with the ‘wicked problems’ identified but there is one, rather large one, missing. It is hinted at throughout and is reflected in some of the others and that is that conservation, as embodied in Canadian law and imagination, is about preserving a people free wilderness and IPCAs are not. Thus, what is required is a profound shift in how the Canadian state, its conservation organizations and its public think about conservation. Rather wicked, I would say..

So overall I am inclined to really like the article, I find it well written and relatively well organized. But to the uninitiated, it might be confusing (eg do most people understand what is being said when the authors discuss the Canadian constitution and the difference between treaties/untreatied?). And each of the wicked problems could be better substantiated in the literature. My concern is that for those skeptical of the ability of IPCAs to deliver on conservation, this article doesn't do enough to garner their support. What it does do is, for those in Canada, identify some sticky points as IPCAs gain traction. Many of those sticky points are already outlined in the ICE report (thought the ICE report is necessary positive and does not discuss Canada's motivations or risks per se). I think, at least, more work should be done at the end to return to the international situation and distill the lessons from Canada for the international push towards the Aichi targets.I would also like to hear more of a first person voice in that it helps situate the authors vis-a-vis the process they describe and would make the intent of the article clearer. It may not be a good fit for Land, I'll allow you to decide that.

-quotes from UNDRIP on p 12 require page numbers or at least article numbers.

Author Response

Thank you for asking me to review this manuscript. I am fully behind the authors project and applaud the authors for identifying an exciting moment in Canadian conservation and advancing an analysis of the challenges, or ‘wicked problems’ as they see them.  I do, of

course, have some concerns and suggestions for improvement.

Response: We would like to thank this reviewer for their time and feedback. We have done our best to respond to each of the suggestions.

1. While I recognize the time constraints inherent in trying to put forward some analysis in a timely manner, I have some concerns about how it is done here. Mostly a purely literature based analysis and review on such a new and emerging process is bound to miss some key points. At the very least, a more thorough review of the popular coverage of Pathway 1 needs to be incorporated, if not interviews with key Indigenous 'insiders'. In an article so clearly aligned with a desire to see Indigenous-led conservation take root in Canada, I would have expected to hear more Indigenous voices. While co-authorship is not always possible, a few key-informant interviews would have added some nuance to the account. Pathway 1 and its implementation is a live act and having someone from 'behind the scenes' would have been beneficial. At the very least, there are quite a few popular pieces containing quotations from members of ICE or key members of the Indigenous leadership on this issue, which should have been at least cited. There is also a fair few allied organizations with important analysis the authors could have built upon. Here are a few I would recommend incorporating (there are others):

Indigenous scholars whose writing would support some of what you put forward (in terms of nation-to-nation relations):

N. Reo, K. Whyte, D. McGregor, M. (Peggy) Smith, and J. Jenkins, “Factors that support

Indigenous involvement in multi-actor environmental stewardship,” AlterNative, vol. 13, no. 2, pp. 58–68, 2017.

Allied scholars involved in supporting ICE:

F. Moola and R. Roth 2018. “Moving beyond colonial conservation models: Indigenous Protected and Conserved Areas offer hope for biodiversity and advancing reconciliation in the Canadian boreal forest.,” Environ. Rev.https://doi.org/10.1139/er-2018-0091

Allied organizations (WCEL) on legal frameworks:

G. Lloyd-Smith, “Indigenous protected areas gaining momentum - but are they recognized by law?",” Environ. Law Alert Blog, 2017.

David Suzuki Foundation – a couple of reports on Tribal Parks and IPCAs in BC

https://davidsuzuki.org/science-learning-centre-article/let-us-teach-you-exploring-empowerment-for-indigenous-protected-and-conserved-areas-in-b-c/

Indigenous leaders, including member of ICE:

V. Courtois and S. Nitah, “Indigenous-led conservation offers a path to global leadership and reconciliation,” The Star, 23-Jan-2018.

E. Enns, “Tribal Parks in Canada: Modern expressions of Indigenous Governance systems,” ICCA Consortium Blog, 2014. [Online]. Available:

https://www.iccaconsortium.org/index.php/2014/10/05/tribal-parks-in-canada-modernexpressions- of-indigenous-governance-systems/.

Response: Our article aims to problematize the issues around the development IPCAs in Canada. For this reason, our paper is more conceptual and based on recent policy developments. We leave the category of our paper in Land to the discretion of the editors (could be “Article” or “Review”). We agree that this article would benefit from the inclusion of Indigenous voices through popular sources. We have now included quotes from ICE members and allies. We appreciate the reviewer pointing us to sources, and have included key points and references to them.

 2. I also have some concerns about fit and scoping. I am not 100% convinced Land is the best place for this piece. I leave it up to the authors and editors to decide precisely on this issue but let me explain my reasoning. A) I think an allied Canadian audience (such as myself) will enjoy this paper as it discusses a very interesting development in Canadian conservation but it takes too much for granted for a non-Canadian audience or for a Canadian audience who isn’t already fairly well versed in reconciliation, nation-to- nation relationships and the like. This could be resolved with a clearer framing and some footnotes defining terms. What exactly should someone from outside of Canada take away from this? Is there a lesson for other settler colonial states? Is there a lesson about Aichi targets? If the audience is Canadian, perhaps a regional journal would be a better fit. Or perhaps a journal with a readership more familiar with Indigenous people and conservation? B) This is not a research article and contains no data. It is much more of a paper serving to identify research needs or, at least, challenges to actualizing IPCAs in Canada. Perhaps that’s ok for Land, not sure as I am unfamiliar with the journal.

Response: A) We believe that national approaches to grappling with Indigenous and IPCA issues associated with achieving Aichi targets are important contributions to the international literature on biodiversity conservation and protected areas (theme of the journal’s special call). We have added detail to the discussion on reconciliation and nation-to-nation relationships and other concepts that would be less familiar to international audiences. We have also added language to the final section (Conclusions) to situate the research within the international scope for Indigenous protected areas.  We have added an explicit indication in the abstract, introduction and conclusion that several of the issues we discuss with reference to Canada are of potentially important relevance to other colonial nation states grappling with similar challenges related to IPCAs and international targets.

B) The article does not include primary data, however, it is “original research” and presents a “substantial amount of new information”, as per Land guidelines. As mentioned earlier, we are open to this article being published as a Review if the editors of Land feel that this category is a better fit.

3. I am pleased to see the authors identify some of the risks in Pathway 1 and I agree it is necessary to be a bit cautious about Canada’s motivations – there is a very real possibility that a focus on quantitative targets marginalizes many if not most Indigenous nations and replicates colonial exclusions (had the authors done some interviews, they would have been able to identify a few ways this is already happening.) I agree with the ‘wicked problems’ identified but there is one, rather large one, missing. It is hinted at throughout and is reflected in some of the others and that is that conservation, as embodied in Canadian law and imagination, is about preserving a people free wilderness and IPCAs are not. Thus, what is required is a profound shift in how the Canadian state, its conservation organizations and its public think about conservation. Rather wicked, I would say.. 

Response: We agree that preserving a people free wilderness is “wicked problem” and unpacked this further in Section 3.1 – Canada’s Protected Areas: the Colonial Past and Present and have added a new sub-section (5.1). We have also added quotes and references to illustrate that existing Canadian initiatives for conservation can and do marginalize Indigenous nations and replicate colonial exclusions.

So overall I am inclined to really like the article, I find it well written and relatively well organized. But to the uninitiated, it might be confusing (eg do most people understand what is being said when the authors discuss the Canadian constitution and the difference between treaties/untreatied?). And each of the wicked problems could be better substantiated in the literature. My concern is that for those skeptical of the ability of IPCAs to deliver on conservation, this article doesn't do enough to garner their support. What it does do is, for those in Canada, identify some sticky points as IPCAs gain traction. Many of those sticky points are already outlined in the ICE report (thought the ICE report is necessary positive and does not discuss Canada's motivations or risks per se). I think, at least, more work should be done at the end to return to the international situation and distill the lessons from Canada for the international push towards the Aichi targets. I would also like to hear more of a first person voice in that it helps situate the authors vis-a-vis the process they describe and would make the intent of the article clearer. It may not be a good fit for Land, I'll allow you to decide that. 

Response: We have added footnotes to provide further explanation for those who might not be familiar with Canadian contexts. We have added quotes and references to support the ability of IPCAs to deliver on conservation. As suggested, we have also returned to the international context in our conclusions (as also noted above).

-quotes from UNDRIP on p 12 require page numbers or at least article numbers.

Response: We have now added the page numbers and text to make more clear.

Round 2

Reviewer 2 Report

I am pleased to recommend this paper be published as part of a special issue. The revisions adequately address my concerns and make the paper more useful for an international readership. I also believe the inclusion of more quotations from some of the key actors on ICE greatly improves the manuscript.

Author Response

Thank you for your constructive feedback. We appreciate the time and consideration that you have give to our work.